# Review of Wind Tunnel Modelling of Flow and Pollutant Dispersion within and from Naturally Ventilated Livestock Buildings

**Štěpán Nosek** [1,*], **Zbyněk Jaňour** [1], **David Janke** [2], **Qianying Yi** [2], **André Aarnink** [3], **Salvador Calvet** [4], **Mélynda Hassouna** [5], **Michala Jakubcová** [1], **Peter Demeyer** [6] **and Guoqiang Zhang** [7]

1   Czech Academy of Sciences, 182 00 Prague, Czech Republic; janour@it.cas.cz (Z.J.); michala@it.cas.cz (M.J.)
2   Department of Engineering for Livestock Management, Leibniz Institute for Agricultural Engineering and Bioeconomy (ATB), Max-Eyth-Allee 100, 14469 Potsdam, Germany; djanke@atb-potsdam.de (D.J.); QYi@atb-potsdam.de (Q.Y.)
3   Wageningen Livestock Research, 6708 WD Wageningen, The Netherlands; andre.aarnink@wur.nl
4   Institute of Animal Science and Technology, Universitat Politècnica de València, Camino de Vera s.n., 46022 Valencia, Spain; salcalsa@upvnet.upv.es
5   French National Institute for Agricultural Research, CEDEX, F-35042 Rennes, France; hassouna@rennes.inra.fr
6   Technology and Food Science Unit-Agricultural Engineering, Institute for Agricultural and Fisheries Research (ILVO), Burg. Van Gansberghelaan 115 Bus 1, 9820 Merelbeke, Belgium; peter.demeyer@ilvo.vlaanderen.be
7   Department of Civil and Architectural Engineering—Design and Construction, Aarhus University, 8000 Aarhus C, Denmark; guoqiang.zhang@eng.au.dk
*   Correspondence: nosek@it.cas.cz; Tel.: +420-266-053-382

**Abstract:** Ammonia emissions from naturally ventilated livestock buildings (NVLBs) pose a serious environmental problem. However, the mechanisms that control these emissions are still not fully understood. One promising method for understanding these mechanisms is physical modelling in wind tunnels. This paper reviews studies that have used this method to investigate flow or pollutant dispersion within or from NVLBs. The review indicates the importance of wind tunnels for understanding the flow and pollutant dispersion processes within and from NVLBs. However, most studies have investigated the flow, while only few studies have focused on pollutant dispersion. Furthermore, only few studies have simulated all the essential parameters of the approaching boundary layer. Therefore, this paper discusses these shortcomings and provides tips and recommendations for further research in this respect.

**Keywords:** ammonia; GHG; atmospheric boundary layer; livestock building; natural ventilation; pollutant dispersion

## 1. Introduction

Excess ammonia emissions into the atmosphere pose a serious environmental issue. Gaseous ammonia reacts with other atmospheric species and transforms into fine particulate matter ($PM_{2.5}$), such as ammonium nitrate and ammonium sulphate. These fine particles can travel as far as 2500 km [1] due to their low deposition velocities and the characteristics of the atmospheric boundary layer (ABL), such as its mean velocity and turbulence intensity. While the wet deposition of ammonia leads to soil acidification and eutrophication [2], its dry deposition causes visible foliar damage [1]. Higher ammonia concentrations inside livestock buildings can harm the health of both animals and farmers.

In 2016, the agricultural sector in the European Union was responsible for 92% of the total ammonia emissions [3]. The primary agricultural sources of these emissions were livestock buildings, feedlots, and manure storage and fields.

One of the starting points of the formation of ammonia is the floor of livestock buildings, where the excrements of animals are deposited. Here, the ammonia emission (volatili-

sation from the manure surface) is driven by the local microenvironment conditions above the manure surface, such as the gaseous ammonia concentration, the mean air velocity, turbulence intensity, and air temperature. The properties of the manure (e.g., the concentration of urea, pH, and temperature) have a critical impact on the ammonia emission [4,5]. According to the first Fick's law, the ammonia emission rate decreases with the increase of the ammonia concentration in the air or a decrease in the ammonia concentration in the manure. In contrast, previous studies on ammonia mass transfer above the emission surface have shown that the mean air velocity and turbulence [6,7] and the temperature difference between the manure and the surrounding air increase the emission [8]; the higher the airflow and turbulence intensity above the manure surface, the higher the ammonia emission. In the case of air temperature, the temperature difference between the manure and the surrounding air is important: The higher the temperature difference, the higher the ammonia emission from the manure. Simple models have been developed to predict the ammonia emission from naturally ventilated livestock buildings (NVLBs) based, for example, on the outdoor temperature [9]. However, detailed information about the ammonia emission mechanisms is hidden by such models.

The modelling of ammonia dispersion within and from NVLBs is more challenging than in cases of those buildings that are ventilated mechanically (e.g., piggery and poultry houses) [10,11]. Compared to mechanically ventilated livestock buildings, NVLBs must have larger openings in order to provide enough clean air and remove excess heat. However, due to these large openings, the wind directly impacts the indoor environment of NVLBs. During the warm and transitional seasons, the openings of the NVLB represent more than 50–100% of the building wall area. Hence, the external flow sweeps with ease through the building and may increase ammonia emissions, due to the higher airspeed and turbulence intensity above the manure surface. Even during winter conditions, the openings are still large enough to allow interaction between the external flow and the indoor environment, which can produce complex turbulent flow patterns inside and around the building (see Figure 1).

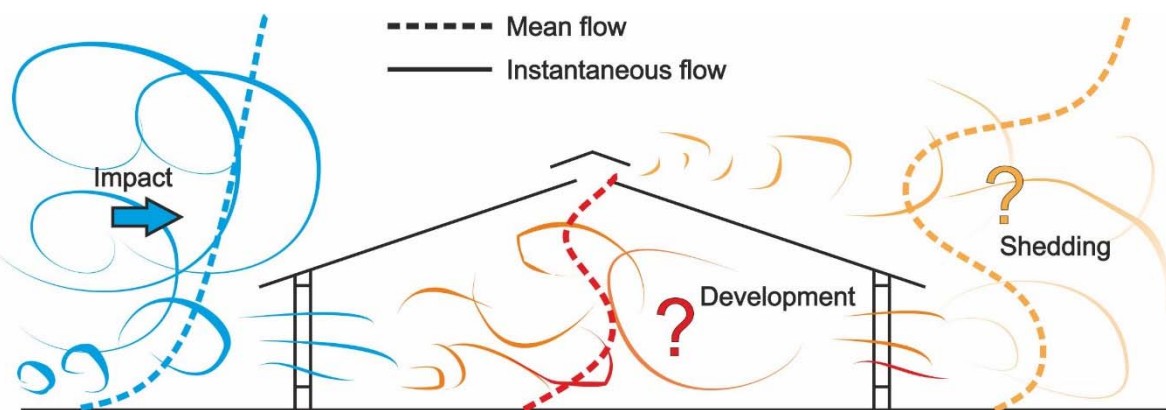

**Figure 1.** Sketch of possible instantaneous (solid lines) and mean (dashed lines) flow characteristics of the turbulent flow impacting a naturally ventilated livestock building (NVLB) (blue), developed within the NVLB (red), and shed by the NVLB geometry (orange). The question marks represent the unknown ammonia emission and dispersion. The flow is from left to right.

Owing to the diurnal weather cycles and surrounding topology, the wind is highly turbulent. The instantaneous intensity and direction of the wind measured at the building height rarely correlate with those of the flow measured at the building's openings. Even if the time is averaged over, for example, 30 min, better results may not be obtained, as the turbulent structures developed naturally in the incoming flow have a bigger time scale than these averages [12]. Thus, to obtain representative (i.e., statistically steady) data that relate the flow within the building with the outdoor wind, one needs to perform air velocity measurements under steady conditions and for a sufficiently long time. However, steady

conditions are rare due to the atmospheric turbulence and, hence, longer time-averaging does not solve the issue [12]. Ogink et al. [13] pointed out a typical example of large scatter (by a factor of 2) in reported ammonia emissions of dairy cattle housings, where it was unclear whether the spread could be attributed to the effects of climatic conditions, housing layout, or different measurement concepts.

Two research methods are available to improve the understanding of airflow and ammonia dispersion within and from an NVLB, concerning the in situ experiments mentioned above: computational fluid dynamics (CFD) and wind tunnel simulations. CFD is a numerical framework that can solve all processes (i.e., mechanical, thermodynamic, and chemical) of ammonia emission. However, the trustworthiness of the results needs to be established through proper validation. The validation process is even more crucial for predicting flows and pollutant dispersions around bluff bodies with sharp edges (the flow around NVLBs is an excellent example), where unsteady turbulent vortices of different length and timescales are shed into the surrounding flow [14,15]. This shedding prediction is the central issue of most CFD turbulent models based on the so-called Reynolds-averaged Navier–Stokes equations (RANS), as unsteady fluctuations cannot be reproduced by these models [15]. Indeed, there exist CFD models, such as large-eddy simulations (LES) and direct numerical simulations (DNS), that can tackle this limitation by resolving the turbulent flow either through a filtered grid (LES) or without any filtering (DNS). However, these resolving models demand extraordinary computational effort and are, therefore, still object to more or less basic academic purposes [16].

On the contrary, wind tunnel simulations employ real fluids, real pollutants, and real geometries (though at a reduced scale). The conditions can be fully controlled and the specific parameters can be adjusted independently. Measurement techniques, such as time-resolved particle image velocimetry (TR-PIV) and planar laser-induced fluorescence (PLIF), can provide data with high spatiotemporal resolution and representativeness. Such data are vital for the validation and tuning of RANS models. Compared to in situ experiments, physical modelling in wind tunnels is more favourable in terms of costs. Time is also on the side of wind tunnel modelling. For instance, a reduction of the modelled case by 100 means that measuring 1 min in the wind tunnel corresponds to 100 min in situ. However, as mentioned above, the conditions during in situ measurements are rarely steady during these 100 min, and hence, insufficient representative data are usually obtained [11,17].

Since the 1980s, wind tunnels have been used to understand the flow and dispersion processes inside NVLBs, not just due to the advantages mentioned above. Still, the physical modelling of such reduced-scale flows has limitations, and the obtained results must be interpreted accordingly. Therefore, this paper seeks to review previous wind tunnel studies that have addressed the natural ventilation of livestock buildings, as well as to outline the limitations and future improvements in this respect. It should be noted that the ammonia emission from an NVLB is also a specific factor related to animal production and activity within the building. However, these specifics fall outside the scope of the paper. The paper is structured as follows: Section 2 provides an overview of the issues related to the physical modelling of natural ventilation and dispersion in wind tunnels, while tips and recommendations to achieve proper reduced-scale flows in wind tunnels are given. Section 3 reviews previous wind tunnel studies on NVLBs. Finally, Section 4 provides our discussion and conclusions, along with future perspectives on the wind tunnel modelling of flow and pollutant dispersion within and from NVLBs.

## 2. Issues with Wind Tunnel Modelling of Flow and Dispersion Processes in NVLBs

The physical modelling of flow and dispersion processes within an NVLB requires both a properly modelled ABL (into which the building is immersed) and indoor flow. If we restrict ourselves to a simple isothermal case, only two Reynolds numbers (Re) should be fulfilled: external (i.e., related to the ABL, where the representative length and velocity are the ABL height and the mean velocity at that height, respectively) and internal (i.e., associated with the indoor flow). Cermak et al. [18] demonstrated, on a civil building

model with a scale of 1:25, that the internal Re (based on the mean velocity at the opening and the smallest dimension of the room) should be at least $2 \times 10^4$ in order to ensure the independence of the internal flow on the internal Re. Such high Re might be reasonably achieved for livestock buildings due to their large openings (Figure 2). However, special attention should be paid to the modelled scale of turbulence in wind tunnels. Dean [19] demonstrated that even though he modelled the mean velocity and turbulence intensity of the required ABL in the wind tunnel properly, a plume coming from a stack was three times broader than that modelled by Snyder and Lawson [20]. After proper modelling of turbulence spectra (in a way that the spectra modelled in the wind tunnel correspond to those observed at full scale), Dean achieved the same plume breadth, within the limits of experimental uncertainties. The same issue might be expected in the case of modelling the dispersion process within an NVLB. Katayama et al. [21] compared the velocity fields within naturally ventilated rooms of apartment houses obtained from wind tunnel tests with those obtained from full-scale measurements (Figure 3). Their comparison showed that the modelling of the critical ABL characteristics, such as the vertical profiles of the mean velocity, the intensity of turbulence, and vertical momentum fluxes, in the wind tunnel produced similar internal flow fields.

The simulation of non-isothermal flows (e.g., unstable ABLs or convective flows within NVLBs) in wind tunnels is challenging, as both the Reynolds number and the Froude number (Fr) should be fulfilled. The latter represents the ratio of internal forces (flows) to buoyancy forces (flows). The simulation of the dominance of convective flows above the inertial means that the Fr is lower than unity. Attaining such a low Fr entails increasing the difference between the approaching flow temperature and, for example, the wind tunnel floor in order to produce sufficiently high heat flux. However, due to technical constraints, this high flux is usually achieved under the sacrifice of lowering the internal forces—the wind tunnel wind speed. Hence, the Re independence is usually relaxed. Several wind studies have attempted to tackle this problem concerning environmental flows [22–25]. However, the simulation of stable or unstable flows to study their impact on the dispersion within and from NVLBs is more challenging due to the additional internal Re independence. Indeed, there has been no wind tunnel study on such flows concerning the natural ventilation of any building. Even the modelling of isothermal flows in a wind tunnel is not a trivial task. Therefore, in the following sections, we restrict ourselves to the proper modelling of neutrally stratified flows in wind tunnels and discuss under which conditions such modelling can be achieved for the cases of flow and pollutant dispersion within and from NVLBs.

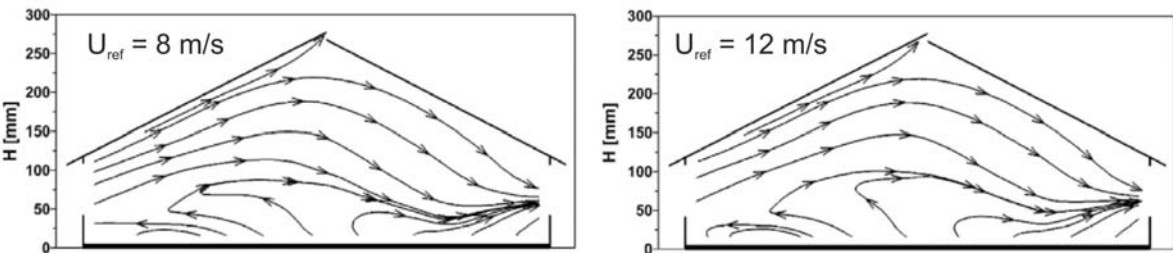

**Figure 2.** Internal Reynolds number independence test within a dairy building for two wind tunnel freestream velocities: 8 m/s (left) and 12 m/s (right). Note that the streamlines (lines with arrows) do not change significantly with the change in the approach wind speed (freestream velocity). Adapted from [26]—Copyright 2018, Elsevier.

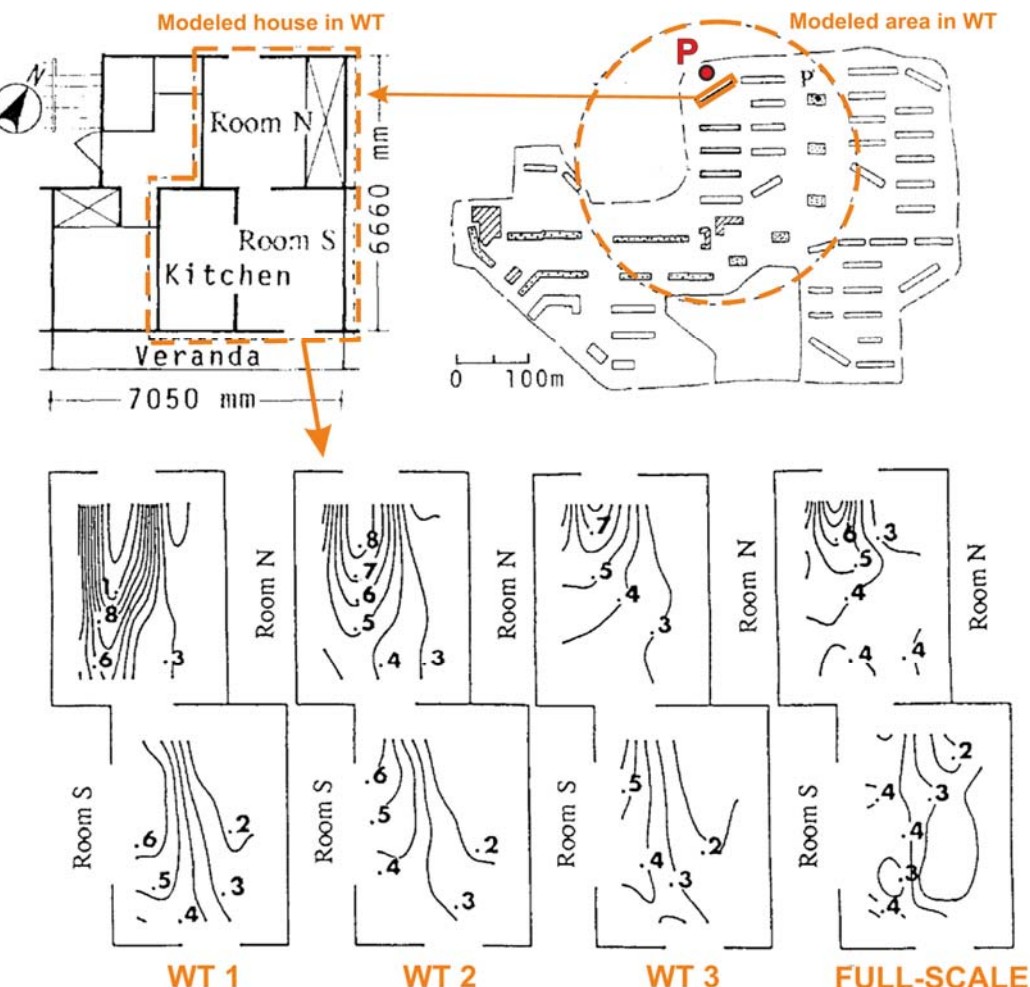

**Figure 3.** Comparison of the velocity fields within the rooms (Room N and Room S) of an apartment house between different wind tunnel simulations (WT 1, 2, 3) and full-scale experiments. WT 1—simulation of simple wind without turbulence generators; WT 2—simulation of wind with turbulence generators; WT 3—full simulation of the appropriate ABL. Point P denotes the position where the wind profiles were obtained. Adapted from [21]—Copyright 1992, Elsevier.

### 2.1. Boundary Conditions

The first process in any wind tunnel study is the setting of appropriate boundary conditions. A wind tunnel with cross-dimensions of 1 m × 1 m will not develop a boundary layer bigger than 1 m. However, one of the critical measures of ABL turbulence, the longitudinal integral length scale ($L_{ux}$), is generally bigger than the depth of the ABL and other turbulent structures (e.g., very large organised structures), which may develop along with the flow. Such wind tunnels can adequately model the vital part of the ABL—the so-called surface layer (SL)—at the scale of 1:100 and higher. However, for a smaller scale ratio (e.g., 1:50), approximately the upper half of the modelled SL will be missing. One needs a wind tunnel of a 2 m × 2 m cross section to properly model the full SL at a scale of 1:50. Due to the Re reduction, both wind tunnels cannot model the smallest full-scale eddies (observed at the so-called Kolmogorov microscale) properly, although a bigger wind tunnel may model the broader spectrum of turbulence. Fortunately, the most energy-containing eddies, which play a crucial role in the dispersion of pollutants within the SL, are those that are much larger (about hundreds and thousands of millimetres at full scale) than those at the Kolmogorov microscale. Thus, dispersion modelling does not suffer too much due to this turbulence spectra reduction [27–29].

Developing appropriate boundary conditions to model the ABL (or SL) in a wind tunnel is not an easy task. Usually, a trial-and-error process is undertaken to match all

essential characteristics (e.g., roughness length, $z_0$; displacement height, $d_0$; friction velocity, $u_*$; integral length scale; and spectrum of turbulence) of the simulated ABL with those recommended by practical guidelines (see, e.g., [30]). To do so, physical modellers have used a well-established combination of vortex generators and roughness elements at the bottom of their wind tunnels. The overall purpose of these devices is to retard the mean flow close to the wind tunnel bottom and induce vorticity turbulence into the boundary layer, due to the relative shortness of the wind tunnel.

For the vortex generators, Irwin's [31] or Counihan's [32] spires are mostly used at the beginning of the wind tunnel development section. At the same time, the roughness elements are represented by various shapes and are usually distributed in squared or staggered form in the remaining part of the development section (Figure 4). The role of the vortex generators is to produce vortices with a vertical axis or large-scale (longitudinal) eddies. Further downstream, these vortices or eddies interact with those induced by roughness elements. To provide an initial momentum deficit, some laboratories have used a castellated barrier [33], or a wall [34] of height smaller than 5% of the wind tunnel size, in front of the vortex generators. The shape of the roughness elements is also critical to reproduce appropriate turbulent structures within a modelled ABL [35,36].

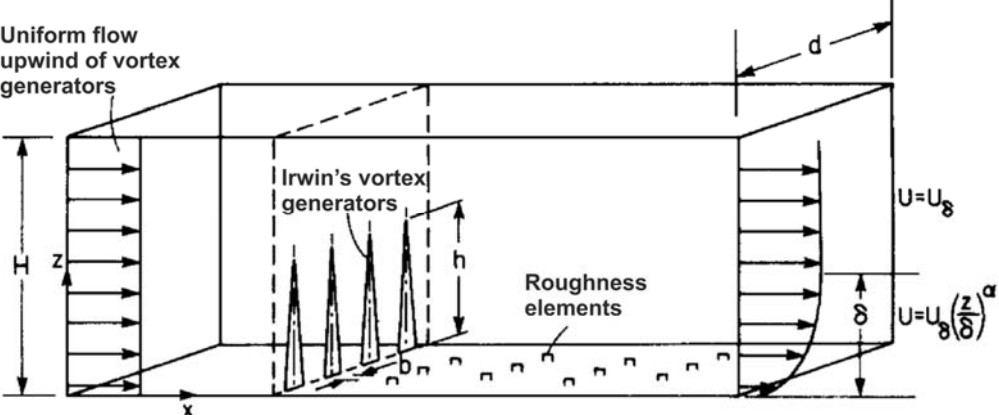

**Figure 4.** The most used devices for development of atmospheric boundary layer (ABL) in wind tunnels including Irwin's spires and roughness elements (adapted from [30]—Copyright 1981, Elsevier).

### 2.2. Finding of ABL Parameters

To find the desirable roughness length, $z_0$, and displacement height, $d_0$, of the modelled ABL, the common procedure of fitting the mean velocity ($U$) profile to the logarithmic law-of-the-wall is used:

$$U = \frac{u_*}{\kappa} ln\left(\frac{z - d_0}{z_0}\right),\tag{1}$$

where $\kappa$ is the von Kármán constant (usually taken as 0.4) and $z$ is the measured height. The velocity profile should be measured at the domain entrance (i.e., at the end of the development section of the wind tunnel). For the estimation of friction velocity, $u_*$, one can use the assumption that the measured mean vertical momentum flux ($< u'w' >$, where $u'$ and $w'$ are the fluctuation components of the longitudinal and vertical velocities, respectively, and the angular brackets represent time averaging) is nearly equal to $u_*^2$ in the region just above the roughness height [37]. Equation (1) can be solved for $z_0$ in cases when $d_0$ is equal to zero (see Table 1) or when $d_0$ is obtained from fitting (using the least squares method) of the measured velocity profile of the modelled ABL to the power ($\alpha$) law:

$$\frac{U(z)}{U_{ref}} = \left(\frac{z - d_0}{z_{ref} - d_0}\right)^{\alpha},\tag{2}$$

where $U_{ref}$ is the velocity at the reference height.

**Table 1.** Roughness lengths ($z_0$), profile exponents ($\alpha$), and zero plane displacements ($d_0$), according to [32].

| Roughness Class | Slightly Rough | Moderately Rough | Rough | Very Rough |
|---|---|---|---|---|
| Type of terrain | Snow, water surface | Grassland, farmland | Park, suburban area | Forest, inner-city area |
| $z_0$ (m) | $10^{-5}$–$5 \times 10^{-3}$ | $5 \times 10^{-3}$–0.1 | 0.1–0.5 | 0.5–2 |
| $\alpha$ | 0.08–0.12 | 0.12–0.18 | 0.18–0.24 | 0.24–0.40 |
| $d_0$ (m) | $\approx 0$ | $\approx 0$ | $\approx 0.75 \times h$ | $\approx 0.75 \times h$ |

$h$ is the mean height of vegetation and buildings (in m).

The longitudinal integral length scale, $L_{ux}$, is an essential measure of the dominant eddy size in a modelled ABL and is usually determined from the autocorrelation of the time series, assuming that Taylor's hypothesis about frozen turbulence is valid. The modelled $L_{ux}$ is then compared with that observed in full-scale measurements for different roughness lengths and heights. Figure 5 reflects the following characteristics of $L_{ux}$ in a full-scale ABL, which were observed by Counihan [28]: $L_{ux}$ is many times bigger than the height at which it is observed; $L_{ux}$ increases with increasing height, while it decreases with increasing roughness; and from a height of around 300 m above the terrain roughness, $L_{ux}$ decreases with height, regardless of terrain roughness. These characteristics are fundamental, as they facilitate the first estimation of the dimensions of the roughness elements and vortex generators used to model a required ABL.

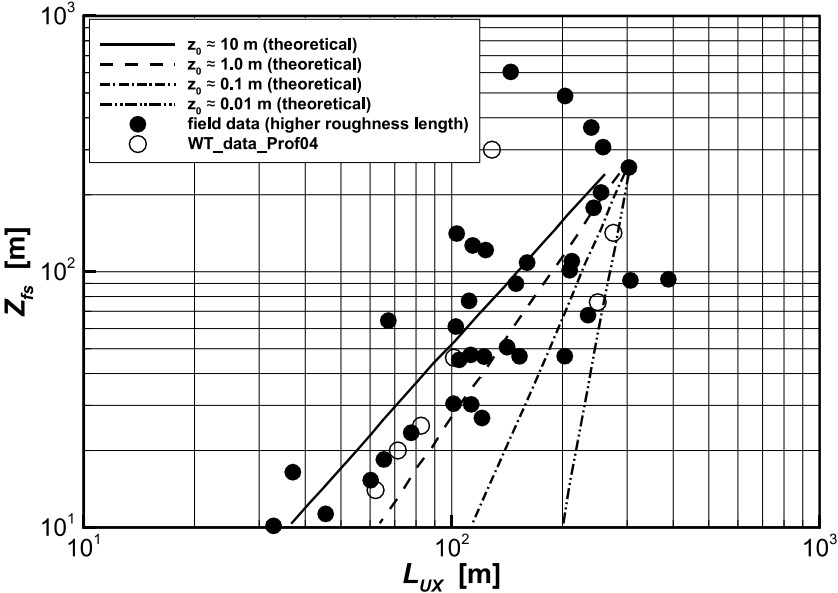

**Figure 5.** Comparison of the longitudinal integral length scales of turbulence, $L_{ux}$, between wind tunnel (open circles, [38]) and full-scale measurements over rough terrain (filled circles, [28]) at different full-scale heights ($Z_{fs}$). The lines represent theoretical values of $L_{ux}$ for an appropriate roughness length $z_0$. Adapted from [38]—Copyright 2016, Springer.

The last important characteristic of the simulated ABL is the spectrum of turbulence (or, more precisely, the spectral distribution of the kinetic energy of turbulence). As in the case of the integral length scale of turbulence, the spectrum of turbulence (obtained from velocity time series using the discrete Fourier transform) should be tested at different heights of the modelled ABL and should be in good agreement with that observed in full-scale measurements. The evaluation of the agreement entails a qualitative comparison (matching) of the curves of the turbulence spectra between the wind tunnel and full-

scale observations. If full-scale data are not available, the spectrum of turbulence can be approximated by

$$\frac{fS_{uu}(f,z)}{\sigma_u^2(z)} = \frac{Af_{red}}{\left(E + B\ f_{red}^C\right)^D}$$

(3)

for the entire micrometeorological frequency range, where $f$ is the frequency of the velocity fluctuations, $S_{uu}$ is the spectral density distribution of the longitudinal velocity, $f_{red}$ is the reduced frequency, A–E are the approximation constants, and $\sigma_u$ is the standard deviation of the fluctuation component of the longitudinal velocity. For the approximation constants and reduced frequency, the most used are those stated by Kaimal et al. [39], Simiu and Scanlan [40], and von Kármán [41], which are listed in Table 2. Note that von Kármán's approach uses, for the reduced frequency, the longitudinal integral length scale of turbulence ($L_{ux}$) instead of the height (z).

**Table 2.** Approximation constants (A–E) and reduced frequency (f$_{red}$), according to Kaimal et al. [39], Simiu and Scanlan [40], and von Kármán [41]. Adapted from [32].

| Approx. Const. | A | B | C | D | E | f$_{red}$ |
|---|---|---|---|---|---|---|
| Kaimal et al. | 16.8 | 33.0 | 1 | 5/3 | 1 | $\frac{fz}{U(z)}$ |
| Simiu and Scanlan | 32.0 | 50.0 | 1 | 5/3 | 1 | $\frac{fz}{U(z)}$ |
| von Kármán | 4.0 | 70.78 | 2 | 5/6 | 1 | $\frac{fL_{ux}(z)}{U(z)}$ |

## 3. Previous Wind Tunnel Studies on NVLBs

In the scientific databases Scopus and Web of Science, we found 19 peer-reviewed articles (see Table 3) that addressed the flow or dispersion processes within or from NVLBs and used wind tunnel methods. Table 3 shows that the most of the previous wind tunnel studies have focused on the investigation of outdoor (16 studies) and indoor (11 studies) flow, while research into either outdoor (4 studies) or indoor (3 studies) pollutant dispersion did not attract significant attention. The main explanation for this might be the difficulties in simulating appropriate pollutant source and concentration measurements, especially within the NVLB. Another critical indicator provided by Table 3 is that there were minimal (only two) wind tunnel studies that simulated the all-important ABL characteristics discussed in Section 2. The following sections overview the parameters and their effects on flow and pollutant dispersion, either within or around an NVLB.

### 3.1. The Flow Studies

#### 3.1.1. The Effect of Openings

One of the first published studies on the simulation of flow within a scale model of an NVLB in a wind tunnel was that of Choiniere et al. [42]. They reported that there was no readily available information on airflow patterns in a naturally ventilated grower–finisher barn for different air inlet designs and windbreak panels. To obtain such information, they conducted smoke visualisation of isothermal airflow patterns for four air inlet types. The barn model was scaled down to 1:20, and the Reynolds number independence (Re > 3500) was observed from qualitative comparisons of flow patterns within the building. Although appropriate boundary conditions (relevant ABL characteristics) were not simulated, the study provided important insight into the impact of air inlets on airflow within an NVLB. For instance, they found that the windbreak panels create extra turbulence, especially for larger openings, which improved air ventilation within the building.

Morsing et al. [43] demonstrated, using a model (1:20) of a finishing pig house with full-length openings at both sidewalls and without a ridge opening, that the height of the wall between the ventilation opening and the ceiling was the most important for the local air velocities within the building.

The impacts of different opening configurations and wind direction on the ventilation of a 1:60 scale model of a dairy cattle barn was studied by De Paepe et al. [44] and De Paepe et al. [45], respectively. Using hot-wire anemometry in their first study, De Paepe et al. [44] observed that enlarging the opening height only at the inlet led to higher velocities at the outlet. They also observed that air velocities at the centre of the house were hardly affected by the inlet opening height. However, without the outlet wall, the velocities at the centre of the house were three to four times higher. In their second study, De Paepe et al. [45] estimated airflow rates through the openings by multiplying the opening's free surface area with the average air velocity measured in the respective ventilation opening. They demonstrated that the estimated airflow rates through the inlet and outlet openings gradually decreased, in both scale models, with increasing wind direction.

Shen et al. [46] focused on the effect of opening configuration on indoor and outdoor airflow conditions and the ventilation rate of naturally ventilated buildings using the laser Doppler anemometry (LDA) and tracer gas methods, respectively. The measured velocity vertical profiles showed that different opening scenarios produce different magnitudes and directions of the streamwise velocity near the building floor. However, the main driver of this velocity behaviour was the inlet opening position, regardless of the outlet openings. If the opening position was near the bottom of the building wall, then a substantial downstream flow was observed near the indoor floor of the building. On the contrary, the upper inlets produce upstream and weaker flow at the floor. However, it was shown that while the location of openings does not have a significant impact on the air change rate, the size of the openings certainly has.

**Table 3.** Related wind tunnel studies on naturally ventilated livestock buildings.

| Study | Poll. Source | 1:x | ABL | Indoor Flow | Outdoor Flow | Indoor Conc. | Outdoor Conc. | Main Findings |
|---|---|---|---|---|---|---|---|---|
| Choiniére et al. [42] | | 20 | | x | x | | | Windbreak panels at the openings improve the pig barn ventilation. |
| Morsing et al. [43] | | 20 | | x | x | | | The position of the ventilation opening is most important for local air velocities in the animal zone. |
| Zhang et al. [47] | Point | 20 | | | x | | x | An obstacle upstream of the barn decreases the contaminant concentration downstream. |
| Ikeguchi et al. [48] | | 20 | Partially | x | x | | | A solid windbreak, as well as a barn, positioned upstream of the investigated barn caused the air to flow towards the windbreak or the upstream barn. |
| Aubrun and Leitl [49] | Point | 400 | Fully | | | | x | The near-field pollutant dispersion from a building is mainly driven by the meandering behaviour of the plume rather than the turbulent diffusion. |
| Ikeguchi et al. [50] | Point | 20 | Partially | x | x | x | x | A contaminant might reach an upwind NVLB, even if it was generated in the downwind NVLB, separated by a distance equal to the NVLB's average height. |
| Sauer et al. [51] | | 300 | Partially | | x | | | Perpendicularly oriented buildings create a large (equal to 10 times the height of the mean building) zone of reduced streamwise velocity and increased turbulence intensity in the wake of the buildings. |
| Hernandez et al. [52] | | 150 | Partially | | x | | | Animal barns themselves had the greatest impact on the flow patterns than vegetative environmental buffers (e.g., trees or shelterbelts). |

**Table 3.** *Cont.*

| Study | Poll. Source | 1:x | ABL | Indoor Flow | Outdoor Flow | Indoor Conc. | Outdoor Conc. | Main Findings |
|-------|------|-----|-----|-------------|--------------|--------------|---------------|---------------|
| De Paepe et al. [44] | | 60 | | x | x | | | Air velocities at the centre of NVLB with constant height outlet openings were hardly affected by change of the inlet opening height. However, without the outlet wall, the velocities were 3–4 times higher. |
| De Paepe et al. [45] | | 60 | | x | x | | | The airflow rates through the inlet and outlet openings gradually decrease with the increase in wind direction. |
| Fiedler et al. [53] | | 60 | | | x | | | Interior details have a significant impact on flow and turbulence inside an NVLB. |
| Ntinas et al. [54] | | 60 | | | x | | | Roof geometry affects the flow downstream but not significantly upstream. |
| Shen et al. [46] | | 25 | | x | x | | | While the location of openings does not have a significant impact on the air change rate of an NVLB, the size of openings certainly has. The outlet openings influence the flow upstream of the building. |
| Konig et al. [55] | area | 200 | Partially | | x | | x | Vertical flow structures produced by both the buildings and the ABL moved contaminated air in higher altitudes, and these were dispersed by higher wind speeds and by these structures. |
| Yi et al. [56] | | 40 | Partially | x | x | | | The discharge coefficient (Cd) of NVLB openings is mainly dependent on the opening size. |
| Yi et al. [26] | | 40 | Partially | x | x | | | With the increase in the opening size, the values of the airspeed and turbulent kinetic energy within an NVLB went up linearly. |
| Nosek et al. [57] | planar | 100 | Fully | x | | x | | The opening size and the type of ABL have a crucial impact on both the flow and the pollutant dispersion within a barn, while the presence of animals and doors openings is insignificant. The pollutant was not well mixed within the barn in any studied case. |
| Yi et al. [58] | | 50 | Partially | | x | | | The roof slope has a significant impact on the wake region—and, hence, on the pollutant dispersion—behind an NVLB, where flow recirculation and higher turbulence intensity occur. |
| Janke et al. [59] | planar | 100 | Partially | x | | x | | The error in the emission estimate from an NVLB could be lowered to less than 5% when the concentrations are measured as a vertical composite sample at the outlet openings. |

The main objective of Yi et al. [26] was to investigate the effects of the sidewall opening configurations on internal airflow fields and air velocity characteristics within the animal-occupied zone using an LDA method. They observed an up-jet airflow pattern within the building when the opening ratio was no greater than 62.71% and was located

beneath the eaves. They also studied the opening configuration where the openings were at the bottom and constituted 44% of the entire sidewalls. In this case, uniform airspeed distributions were observed in the animal-occupied zone. They concluded that care should be taken when using these kinds of opening configurations during extreme cold and windy weather conditions.

Yi at al. [56] investigated the impact of the size of the openings on the discharge coefficient (Cd). They found that the Cd value is mainly dependent on the opening size for large opening configurations and not constant, as has been assumed in previous studies. Therefore, an inaccurate application of constant Cd may lead to severe errors in predicting the airflow rate in NVLBs.

Nosek et al. [57] analysed the evolution of flow patterns within a barn using the TR-PIV method. Their results showed that the sidewall opening height and the type of ABL have a crucial impact on the flow within the barn. The size of the openings also has an impact on the coherent structures of the turbulent flow, which manifested as the well-known Kelvin–Helmholtz instability, due to the developed shear layers at the inlet openings. Larger openings produce more stable turbulent structures within the barn.

### 3.1.2. The Effect of Building Orientation and Shape

Sauer et al. [51] studied the impact of the orientation of swine finisher buildings on the air velocity and turbulence characteristics around these buildings. The buildings were scaled down to 1:300 and used in the following configurations: with one building oriented parallel and perpendicular to airflow and with four buildings oriented parallel, perpendicular, and at a 30° angle to airflow. Their study showed that perpendicularly oriented buildings create a large (equal to 10 times the height of the mean building) zone of reduced streamwise velocity and increased turbulence intensity in the wake of the buildings. The authors suggested that the entrainment and transport of pollutants from manure storage structures may be reduced when these structures are in this zone. However, the authors correctly admitted that the increased turbulence in the wake zone may counter the reduced pollutant emission, and hence, further studies are needed to clarify this effect.

An example of validation of CFD simulations of the flow around different-shaped livestock buildings by data obtained from a wind tunnel experiment can be found in Ntinas et al. [54]. As they used a direct numerical approach for their simulation (where the Navier–Stokes equations were solved directly, without any parametrisation), the validation process had a reciprocal purpose. Indeed, the wind tunnel tests were in good agreement with the numerical simulations. However, a low building Re was modelled ($Re_B = HU/v$ = 1284, where $H$ is the building height and $U$ is the inlet velocity in the wind tunnel or the computational domain). More importantly, different roof geometries and obstacles affected both the instantaneous and time-mean-averaged downstream flow parameters. On the contrary, these parameters did not significantly affect the flow impacting the building.

Yi et al. [58] studied the airflow characteristics downwind of a naturally ventilated pig barn. The changing parameter was the slope (5°, 15°, and 25°) of a roofed outdoor exercise yard. Detailed airflow characteristics downwind of the building were measured by a 2D LDA method. Their results showed that the roof slope has a significant impact on the wake region—and, hence, on the pollutant dispersion—behind the barn, where the flow recirculation and higher turbulence intensity occur. The authors suggested applying other treatment technologies to trap the high-concentrated gaseous pollutants (e.g., odours or ammonia) in this region. Such treatments might contribute to the mitigation of both the pollutant emissions and the burden of the surrounding environment. The authors also pointed out the use of mean air velocity and air turbulence data to validate future CFD studies.

### 3.1.3. The Effect of Obstacles within an NVLB

To obtain insight into the influence of interior equipment on the airflow profile within a naturally ventilated barn, Fiedler et al. [53] chose an approach to produce the approaching flow with turbulence levels as low and uniform as possible (i.e., no spires, turbulence generators, or ground surface roughness were applied) to exclude the impact of ABL turbulence on the indoor flow. They demonstrated that the feeding alley changed the flow pattern and produced higher turbulence within the barn. They concluded that interior details should be considered in the physical or numerical modelling of airflow processes in livestock barns and admitted that the results can only be partially applied in practice.

On the contrary, Nosek et al. [57] demonstrated that the presence of cows (without heat production) has an insignificant effect on the flow within a model of an NVLB when the wind is perpendicular to the sidewall openings. This negligible effect was proven for two simulated ABLs and three sizes of the openings.

### 3.1.4. The Effect of Obstacles Surrounding an NVLB

Hernandez et al. [52] investigated the effect of vegetative environmental buffers (VEB), such as trees and shelterbelts, on the mitigation of odours from swine barns using both wind tunnel experiments and field monitoring studies. The wind tunnel experiments revealed that the animal barns themselves had the most significant impact on the flow patterns. Hence, the previously held assumptions that VEB are the main contributors to lowering the odours around a swine facility were incorrect. However, the authors outlined that more research is needed in order to understand the effects of VEB better further downwind and how VEB intercept and hold odours emitted from livestock buildings.

### 3.2. Dispersion Studies

The effects of an obstacle on airflow and contaminant ($CO_2$) dispersion around a model of NVLB was studied by Zhang et al. [47]. Their results showed that the obstacle decreases the contaminant concentration downstream of the building. Meanwhile, Ikeguchi et al. [48] supported these results by using proposed flow parameters based on the integral values of mean velocity and turbulence intensity, related to airflow momentum in the leeward direction horizontal and vertical plume sizes. Later, Ikeguchi et al. [50] observed that contaminated air might reach an upwind livestock building, even if it was generated in a downwind livestock building. This phenomenon was observed when the buildings were placed at a separation distance equal to their average height.

The study of Aubrun and Leitl [49] first reported fully simulated ABL characteristics in their dispersion study focused on a pig barn. This means that they simulated the velocity and turbulent profiles (including roughness length) and the longitudinal integral length scales and spectrum of turbulence, which corresponded to a rural terrain at a scale of 1:400. The main goal of their study was to simulate the dispersion of odorants from a ventilation system of a pig barn to its surroundings and to investigate the unsteadiness of this dispersion. Using a fast flame-ionisation detector (FFID), they proved that the instantaneous behaviour of the dispersion process could be modelled in a wind tunnel if ABL turbulent properties were carefully replicated. They pointed out the danger of the short averaging times used during in situ experiments, as the scatter in the results could reach more than 100%. The main outcome of their study was the determination of the driving process of the near-field dispersion of odorants from a livestock building. They found that the process is mainly driven by advection (i.e., meandering of the plume) rather than turbulent diffusion. Although they focused on outdoor dispersion, their work demonstrated the importance of wind tunnel studies for studying ABL flow and dispersion phenomena involving livestock buildings.

Konig et al. [55] investigated the mean (advective) pollutant transport from a naturally ventilated barn through a natural barrier filter. They aimed to find the optimal placement and configuration of this barrier in order to reduce the transport and dispersion of ammonia from such a building. They found that vertical flow structures of turbulence produced

by both the buildings and the ABL moved contaminated air (by a passive gas) to higher altitudes, where it was dispersed by higher wind speeds and by these structures. Although the turbulent pollution fluxes were not included and integral length scales and spectra of ABL turbulence were not reported, this study provides important insight into pollutant transport around NVLBs.

The impacts of the ABL, the presence of animals, and the openings' configuration on the pollutant dispersion within a model of a cattle barn was studied by Nosek et al. [57]. While the pollution of the barn was simulated by a ground-level planar source emitting a passive gas at a constant flow rate, the concentration within the building was measured using an FFID. The concentration levels inside the barn decreased proportionally with the increase in the ventilation opening width and, similarly, with a less rough ABL. It was shown that velocity fluctuations are driven by the shear layers of the flow, while the mean concentration gradients drive the concentration fluctuations. The study also demonstrated that the pollutant was not well-mixed within the building in any of the studied cases. A comparison of different methods for identifying the barn ventilation performance showed that a significant underestimation (up to a factor of 5) of the building ventilation may be obtained when using, for example, the ventilation rate index. However, the authors also pointed out the limitations of their study (only the passive pollutant and only one wind direction were simulated). They outlined the need for further investigations which consider these limitations.

Janke et al. [59] systematically investigated the influence of number and position of sensors on the direct estimation of air exchange and pollutant emissions from a scale NVLB. The pollutant source mimicked animal mouth heights in the middle of the barn, where most of the $CO_2$ and methane is released. The concentrations and air velocities were measured at the outlet openings using FFID and LDA probes, respectively. Three configurations of measurement point distributions were tested and were varied by incrementally decreasing the number of lateral sampling positions from 260 to 4. The results showed that a division of the total opening width into three equally long units, where an anemometer is positioned at a height near the horizontal symmetry line, is sufficient to estimate the volume flow rate with an error less than 5%.

## 4. Summary

This review showed the importance of wind tunnels for understanding the flow and pollutant dispersion processes within and from NVLBs. Several studies [49,51,52] have demonstrated the importance of data representativeness, in comparison with those obtained from in situ measurements, and that there is no reason to expect that the flow through the openings is uniform [46,56] or that the pollutant within the building is well mixed even in quasi-stationary cases [57]. The focus of the previous wind tunnel studies was generally on studying the impact of the characteristics of sidewall openings on the flow within the building. Indeed, there is a consensus that the size and position of the sidewall openings are critical parameters for the building's flow behaviour, especially in the animal-occupied zone. In addition, both the opening size [44] and the building geometry [58] have been shown to impact the flow around the building, as well as obstacles surrounding the building [48,50,52]. The effect of obstacles on the flow characterises within an NVLB has attracted less attention as only two studies [53,57] considered this effect during their experiments. While the first study [53] showed that the feeding alley changed the flow pattern and produced higher turbulence within the barn, the second [57] study demonstrated that the presence of cows (without heat production) had an insignificant effect on the flow and ventilation of an NVLB when the wind was perpendicular to the sidewall. Thus, further studies that will consider obstacles within their models of NVLBs as a function of the wind direction are of primary importance. Including heat production into the models of an NVLB will add another valuable, although qualitative, information about the role of non-isothermal effects on the building's flow characteristics or dispersion.

However, from the reviewed papers, two primary deficiencies can be observed. First, there is a lack of studies that have properly modelled the atmospheric boundary layer (ABL) characteristics. It has been shown that due to the large openings of NVLBs, the outdoor (atmospheric) and indoor flows are in mutual interaction and that a better understanding of this interaction will improve the determination of the amount of ammonia emitted from such buildings. As was also demonstrated during this review, the integral length scales and spectrum of turbulence are the ABL parameters that influence the indoor flow appreciably. A wind tunnel study [57] has shown that despite the same mean wind speed at the building height, the flow and pollutant dispersion within the building differed for differently simulated characteristics of the ABL. Therefore, it is recommended to simulate all essential ABL parameters in forthcoming wind tunnel studies intended to model the natural ventilation of livestock buildings. This recommendation is from the perspective of comparability and the standpoint of reliability, which is crucial in CFD validations.

Second, there is a lack of wind tunnel studies that have addressed the pollutant dispersion within or from NVLBs. This lack is a critical finding, as even the dispersion of a passive pollutant cannot be rigorously derived only from flow characteristics. Studies on NVLB [48] and generic building [60] ventilation have demonstrated that the mean concentration gradients are driven by the concentration fluctuations, not by the velocity fluctuations. Other important parameters are the position and type (point, line, or planar) of the source and mechanisms (advective and turbulent) of pollutant transport, which are crucial for understanding the ammonia emission from NVLBs. At the openings and near the bottom of the NVLB, turbulence will play a significant role in ammonia transport. However, there exists no wind tunnel study addressing the turbulent pollution flux (even a passive pollutant), neither near the bottom, nor at the openings of the NVLB. Therefore, future wind tunnel studies that address this issue will be of primary importance for understanding ammonia emission from NVLBs.

Finally, ammonia is not inert, and its emission is strongly related to the spatiotemporal variability of the concentration above the source. Therefore, the central perspective for studying the entire processes related to the ammonia emission from NVLBs is a hybrid approach, which includes both wind tunnel and CFD studies. The power of CFD lies in the possibility of studying all processes coupled with the ammonia emission at all time and spatial scales temporally. The main advantage lies in relating all the essential chemical processes occurring during the emission, dispersion, and deposition of ammonia. There is also no difficulty simulating thermal buoyancy effects or heat productions in an NVLB at relatively high Reynolds numbers. Still, determining a reliable CFD model that will be able to predict all these turbulent flow processes temporally will be a challenging task, for which the use of wind tunnels will undoubtedly play an important role.

**Author Contributions:** Conceptualisation, Š.N. and Z.J.; methodology, Š.N.; validation, Z.J., G.Z., M.H. and P.D.; formal analysis, Š.N.; investigation, Š.N., D.J., Q.Y. and G.Z.; resources, Š.N. and D.J.; data curation, Š.N. and Q.Y.; writing—original draft preparation, Š.N.; writing—review and editing, D.J., M.H., S.C. and Q.Y.; visualisation, Š.N. and Q.Y.; supervision, A.A. and G.Z.; project administration, M.J.; funding acquisition, Š.N. All authors have read and agreed to the published version of the manuscript.

**Funding:** This research was funded by the Ministry of Education, Youth and Sports of the Czech Republic(grant number LTC18070) and by the institutional support RVO: 61388998.

**Institutional Review Board Statement:** Not applicable.

**Informed Consent Statement:** Not applicable.

**Acknowledgments:** We fully acknowledge the European Cooperation in Science and Technology (COST) that supports the cooperation of COST Action LivAGE (CA16106). We also would like to express our thanks to the anonymous reviewers whose constructive comments and helpful suggestions improve the presented paper substantially.

**Conflicts of Interest:** The authors declare no conflict of interest.

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
