# Peer review of "Review of Wind Tunnel Modelling of Flow and Pollutant Dispersion within and from Naturally Ventilated Livestock Buildings"

_applsci, doi:10.3390/app11093783_

Round 1

Reviewer 1 Report

Major Comments

  1. Figure 1 – What is the basis for drawing the lines in this sketch? Is it based on the normal flow characteristics in a similar room?
  2. The authors have articulately organized the review and are easy to follow. I do not have other major comments. However, the author could address a few suggestions for future work to improve the models, perhaps in the summary section.

Minor Comments

  1. Figure 1 Caption – Add the expansion of the acronym NVLB.
  2. Line 64 – it is either “in predicting” or “to predict”. In this case, to predict sounds clear.

Reviewer 2 Report

Publication “Review of wind tunnel modeling of flow and pollutant dispersion within and from naturally ventilated livestock buildings" shows a wide review of the available literature in this area. The described physic-chemical processes in the atmosphere are clear and correctly made. The formulas, calculation formulas and simulations indicate the essence of wind tunnel modeling and pollutant dispersion.

At the end of the article (conclusions), a few specific conclusions resulting from this analysis of the research topic are missing.

After completing them, the article is suitable for publication in the MDPI.
